# Nonequilibrium brain dynamics elicited as the origin of perturbative complexity

**Wiep Stikvoort**[1]*, **Eider Pérez-Ordoyo**[1], **Iván Mindlin**[2], **Anira Escrichs**[1], **Jacobo D. Sitt**[2], **Morten L. Kringelbach**[3,4,5], **Gustavo Deco**[1,6]☯, **Yonatan Sanz Perl**[1,7,8]☯*

**1** Center for Brain and Cognition, Department of Information Technologies and Communications (DTIC), Universitat Pompeu Fabra, Barcelona, Spain, **2** Paris Brain Institute, ICM, Inserm, CNRS, Sorbonne Université, Paris, France, **3** Department of Psychiatry, University of Oxford, Oxford, United Kingdom, **4** Centre for Eudaimonia and Human Flourishing, Linacre College, University of Oxford, Oxford, United Kingdom, **5** Centre for Music in the Brain, Aarhus University, Aarhus, Denmark, **6** Institució Catalana de la Recerca i Estudis Avançats (ICREA), Barcelona, Spain, **7** National Scientific and Technical Research Council (CONICET), Ciudad Autónoma de Buenos Aires , Argentina, **8** Departamento de Matemática y Ciencias, Universidad de San Andrés, Buenos Aires, Argentina

☯ These authors contributed equally to this work.
\* wiep.stikvoort@upf.edu (WS); yonatan.sanz@upf.edu (YSP)

## Abstract

Assessing someone's level of consciousness is a complex matter, and attempts have been made to aid clinicians in these assessments through metrics based on neuroimaging data. Many studies have empirically investigated measures related to the complexity elicited after the brain is stimulated to quantify the level of consciousness across different states. Here we hypothesized that the level of non-equilibrium dynamics of the unperturbed brain already contains the information needed to know how the system will react to an external stimulus. We created personalized whole-brain models fitted to resting state fMRI data recorded in participants in altered states of consciousness (e.g., deep sleep, disorders of consciousness) to infer the effective connections underlying their brain dynamics. We then measured the out-of-equilibrium nature of the unperturbed brain by evaluating the level of asymmetry of the inferred connectivity, the time irreversibility in each model and compared this with the elicited complexity generated after *in silico* perturbations, using a simulated fMRI-based version of the Perturbational Complexity Index, a measure that has been shown to distinguish different levels of consciousness in *in vivo* settings. Crucially, we found that states of consciousness involving lower arousal and/or lower awareness had a lower level of asymmetry in their effective connectivities, a lower level of irreversibility in their simulated dynamics, and a lower complexity compared to control subjects. We show that the asymmetry in the underlying connections drives the nonequilibrium state of the system and in turn the differences in complexity as a response to the external stimuli.

**Data availability statement:** Due to the sensitive nature of clinical data, the Disorders of Consciousness dataset is available upon consultation with Société de Réanimation de Langue Française (https://www.srlf.org/, reference: M-neuro-Doc): secretariat@srlf.org. The Sleep data set is publicly available at https://github.com/yonisanzperl/Perturbation_in_dynamical_models. Code underlying the results in this paper is available at https://github.com/wiepstikvoort/Nonequilibrium-asymmetry-PCI.

**Funding:** W.S. is an FI fellow with the support of AGAUR, Generalitat de Catalunya and Fondo Social Europeo (2022 FI_B 00152, https://agaur.gencat.cat/ca/inici). E.P. is an FPI fellow funded by the Spanish "Ministerio de Ciencia, Innovación y Universidades" (MICIU/AEI/10.13039/501100011033) and "ESF investing in your future" under the grant PRE2020-0961 (https://www.ciencia.gob.es/, https://european-social-fund-plus.ec.europa.eu/en). I.M. is funded by FLAG-ERA research funding organisation (project ModelDXConsciousness, https://www.flagera.eu/). G.D. and A.E. were supported by the Grant PID2022-136216NB-I00 funded by MICIU/AEI/10.13039/501100011033 and by "ERDF A way of making Europe," ERDF, EU (https://ec.europa.eu/regional_policy/funding/erdf_en). G.D. and Y.S.P. were supported by the project NEurological MEchanismS of Injury, and Sleep-like cellular dynamics (NEMESIS) (ref. 101071900) funded by the EU ERC Synergy Horizon Europe (https://erc.europa.eu/apply-grant/synergy-grant). A.E. was also supported by the project eBRAIN-Health - Actionable Multilevel Health Data (id 101058516), funded by the EU Horizon Europe (https://cordis.europa.eu/project/id/101058516). G.D. is also supported by AGAUR research support grant (2021 SGR 00917) funded by the Department of Research and Universities of the Generalitat of Catalunya (https://agaur.gencat.cat/ca/inici). M.L.K. is supported by the Centre for Eudaimonia and Human Flourishing (funded by the Pettit and Carlsberg Foundations) and Center for Music in the Brain (funded by the Danish National Research Foundation, DNRF117, https://www.carlsbergfondet.dk/en/what-we-have-funded/cf20-0698/, https://dg.dk/en/). J.D.S. is supported by the EU ERA PerMed Joint Translational 2019 project (project PerBrain) and by the JTC-HBP project

## Author summary

The Perturbational Complexity Index (PCI) is a measure that was created to distinguish different states of consciousness. By introducing a perturbation in the brain, the brain's response can be compared to its resting state dynamics. It has been shown that the PCI makes this distinction with high accuracy in altered states of consciousness such as sleep, anesthesia, and disorders of consciousness. In this work we looked at what causes the difference in this measure for different states. We used fMRI data from people in wakefulness and deep sleep, and disorders of consciousness. By using computational models and simulating the dynamics of each subject from the dataset, we were able to investigate the relationship between the simulated fMRI-based PCI (sfPCI) values and the underlying dynamics of each model by calculating the asymmetry of connections between brain regions. This asymmetry is known to cause hierarchy and to be present in healthy awake subjects. Here we found that the asymmetry in connections drives the dynamics into an out-of-equilibrium state, which in turn causes the difference in sfPCI values. We conclude that the information to know how the brain will react to such stimulations could already be found in the unperturbed dynamics.

## Introduction

There is an ongoing debate in neuroscience about the connection between consciousness and brain complexity [1]. In this case complexity is defined as the combined presence of integration and segregation and can be measured by computing the spatiotemporal complexity of brain activity [2] and emerges from the underlying critical dynamics [3]. An established approach to investigate this relationship was proposed by Massimini and colleagues, who assessed the perturbation-elicited changes in global brain activity during different states of consciousness, such as wakefulness, sleep, anaesthesia, and disorders of consciousness (DoC) [4–6]. Specifically, they computed the perturbational complexity index (PCI), which captures the significant differences in brain-wide spatiotemporal propagation of external stimulation, and they proposed it as an index of consciousness [4]. Importantly, the PCI had high accuracy in distinguishing different levels of consciousness and the results were replicated in multiple studies, showing its sensitivity and usefulness [7–10]. Although the PCI has been proven to be valuable and accurate, it requires an extensive clinical setup which allows for perturbation of the brain with Transcranial Magnetic Stimulation (TMS) while recording brain activity with electroencephalography (EEG). In 2022 the Explainable Consciousness Indicator (ECI) [11] was presented. In this article it was concluded that the different levels of awareness and arousal could already be distinguished between groups based on the unperturbed time series by using explainable deep learning models. In contrast to the work described in this paragraph, here we will be working with simulations. We will be using personalised models based on

MODELDxConsciousness (https://erapermed.isciii.es/joint-transnational-call2019/, https://www.humanbrainproject.eu/en/collaborate-hbp/partnering-projects/modeldxconsciousness/). The funders had no role in study design, data collection and analysis, decision to publish, or preparation of the manuscript.

**Competing interests:** The authors have declared that no competing interests exist.

fMRI data to create *in silico* perturbations by changing the oscillations in the model, representing the TMS experiment, to calculate a simulated fMRI-based PCI (sfPCI).

The concepts of detailed balance and irreversibility from thermodynamics have been applied to assess a system's ability to exhibit dynamic fluctuations between different states. When a system is reversible, it can still change from one state to another, but it is also able to reverse back without net fluxes of energy or activity. Such a system is said to adhere to detailed balance and is in a state of equilibrium. When a system is said to violate detailed balance, it exhibits net fluxes of, e.g., activity, between its configurations, thereby establishing an arrow of time and rendering the system's state irreversible, i.e., a state of nonequilibrium [12]. Studies of spontaneous brain activity in healthy subjects have demonstrated that the brain operates in such a nonequilibrium state [13].

The main goal of the present work is to establish whether PCI, as a marker of consciousness in the human brain, can be associated with the dynamical nature of brain activity before being perturbed. We hypothesize that the level of nonequilibrium dynamics of the unperturbed system determines the response to an external stimulus and in turn the elicited complexity (sfPCI). To this end, we leverage recent works that proposed different theoretically-grounded approaches to measure the level of nonequilibrium at the whole-brain scale, such as entropy production [14], violation of the Fluctuation Dissipation Theorem (FDT) [15] and temporal asymmetry of brain signals [12,14,15]. In particular, FDT provides a direct link between the state of the system and its response to a stimulus following the Onsager derivation [16–18], equalizing the spontaneous fluctuations and the external perturbation [15]. Based on these thermodynamic concepts, previous works have empirically demonstrated that time asymmetry is reduced during deep sleep [19] and in DoC patients [20]. The work using FDT violations also showed that deep sleep and wakefulness are significantly different [21]. Here we focus on the mechanism behind this nonequilibrium nature of brain dynamics [22] and how this property can be related to PCI. By using simulated resting state data, perturbing the system, and looking at the recovery of its dynamics, we can directly relate the level of nonequilibrium to the sfPCI using one and the same simulation.

In this study, we used a whole-brain modelling approach, as opposed to the previously used empirical TMS-EEG setup. We created personalized whole-brain models fitted to functional Magnetic Resonance Imaging (fMRI) data, which allowed us to exhaustively investigate the perturbative *in silico* response of different states of consciousness. Crucially, we introduced asymmetry in the models creating time hierarchy organization and breaking the detailed balance, causing there to be a state of nonequilibrium [22]. By incorporating the time asymmetry of brain dynamics, which reveal the generative underlying mechanism, asymmetry is introduced in the connections, named generative effective connectivity (gEC) [22]. Specifically, this asymmetry refers to the difference between the connections from region A to region B and from region B to region A, creating a bidirectional connectivity. The gEC is responsible for the balance of the causal interaction between brain regions and in turn the level of nonequilibrium in different states of consciousness.

Taking advantage of the thermodynamics framework as a natural way to quantify nonequilibrium of brain activity and its underlying mechanisms [19], we use the asymmetry of the gEC to explore its relationship with nonequilibrium dynamics and *in silico* sfPCI estimates at subject-level for participants in different forms of altered states of consciousness. Crucially, we found that altered states of consciousness generally have a lower level of asymmetry in their generative fitted connectivities and are thus inherently closer to an equilibrium state in terms of thermodynamics. These models fitted to participants residing in altered states of consciousness, related to lower levels of arousal and/or awareness, also have lower sfPCI values, as well as lower levels of nonequilibrium. Through our models we show that the asymmetry in the generative underlying connections is causing the emergent nonequilibrium dynamics and, at the same time, has an effect on the differences in the complexity elicited by perturbations measured with sfPCI.

## Results

### Overview

We used two fMRI datasets with subjects residing in different states of consciousness: a Disorders of Consciousness (DoC) dataset containing data from subjects in a Minimally Conscious State (MCS) (N = 11), an Unresponsive Wakefulness State (UWS) (N = 10), or a control state (CNT) (N = 13), and a sleep dataset containing data from subjects residing in wakefulness (W) and deep sleep (N3) (N = 18). In both datasets the activity was parcelled into 90 Regions of Interest (ROI's), using the AAL parcellation [23]. A schematic overview of the pipeline can be found in Fig 1. The empirical fMRI data is used to calculate the time-lagged covariance, depicted in Fig 1A and 1B, this time-lag ensures that we have a causal inference in the infrastructure of our models. In other words, we included the time-shifted correlations between ROI's from the empirical data, which have different values from node A to node B than from not B to node A, as described in the introduction. This allows the model to have heterogeneous bidirectional connections as it fits to different values for the two connections. Activity is then simulated by representing each ROI as an oscillator, coupled to other oscillators through a connectivity matrix (see Methods). Briefly, the fitting procedure starts with the Structural Connectivity (SC), which in this case is an averaged SC from a healthy group of subjects. We then start the simulations and update the weights of the model by comparing the simulated time-lagged covariances to their equivalents of the empirical data, i.e., the time-lagged covariances that were calculated for each subject individually (DoC dataset) and each state separately (sleep dataset). Fig 1C shows the comparisons between the newly simulated time-lagged covariance with the equivalent empirical values, and updates the weights of the model's connections. The process is repeated until the error stagnates (S1 Fig) and we have generated an effective connectivity matrix, i.e., the gEC of the model. The asymmetry of the fitted model is calculated, of which an intuitive overview is given in Fig 1D. As mentioned, the gEC contains the weights of the bidirectional connections, the infrastructure of the model. The sfPCI and irreversibility are calculated from the simulated data. As shown in Fig 1E one of the nodes in the model is perturbed by changing the value of the bifurcation parameter for a period. This perturbation influences the rest of the model through its connections, creating a pre- and a post-perturbational time series, i.e., a baseline and a response. The irreversibility is calculated on the preperturbational time series.

### gEC asymmetry characterises different states of reduced consciousness

The asymmetry of the personalised generative Effective Connectivity (gEC) matrices was calculated, yielding one value per subject. As can be seen in Fig 2A and 2B the subjects in normal wakeful consciousness were found to have a higher asymmetry in their networks. To calculate whether the distributions are significantly different from each other we performed a non-parametric permutation test (Wilxocon ranksum test), with a Benjamini-Hochberg correction for multiple comparisons for the DoC dataset, and we tested the effect size using Cohen's d [20]. For the Disorders of Consciousness (DoC) dataset the distribution of the models fitted to the CNT subjects (M = 1053, SD = 217) has significantly higher asymmetry values (p = 0.0043, Cohen's d = 1.34) than the distribution of models fitted to the UWS subjects (M = 650, SD = 329). Between the MCS (M = 831, SD = 383) and UWS subjects (p = 0.0564, Cohen's d = 0.44) and between the MCS and CNT

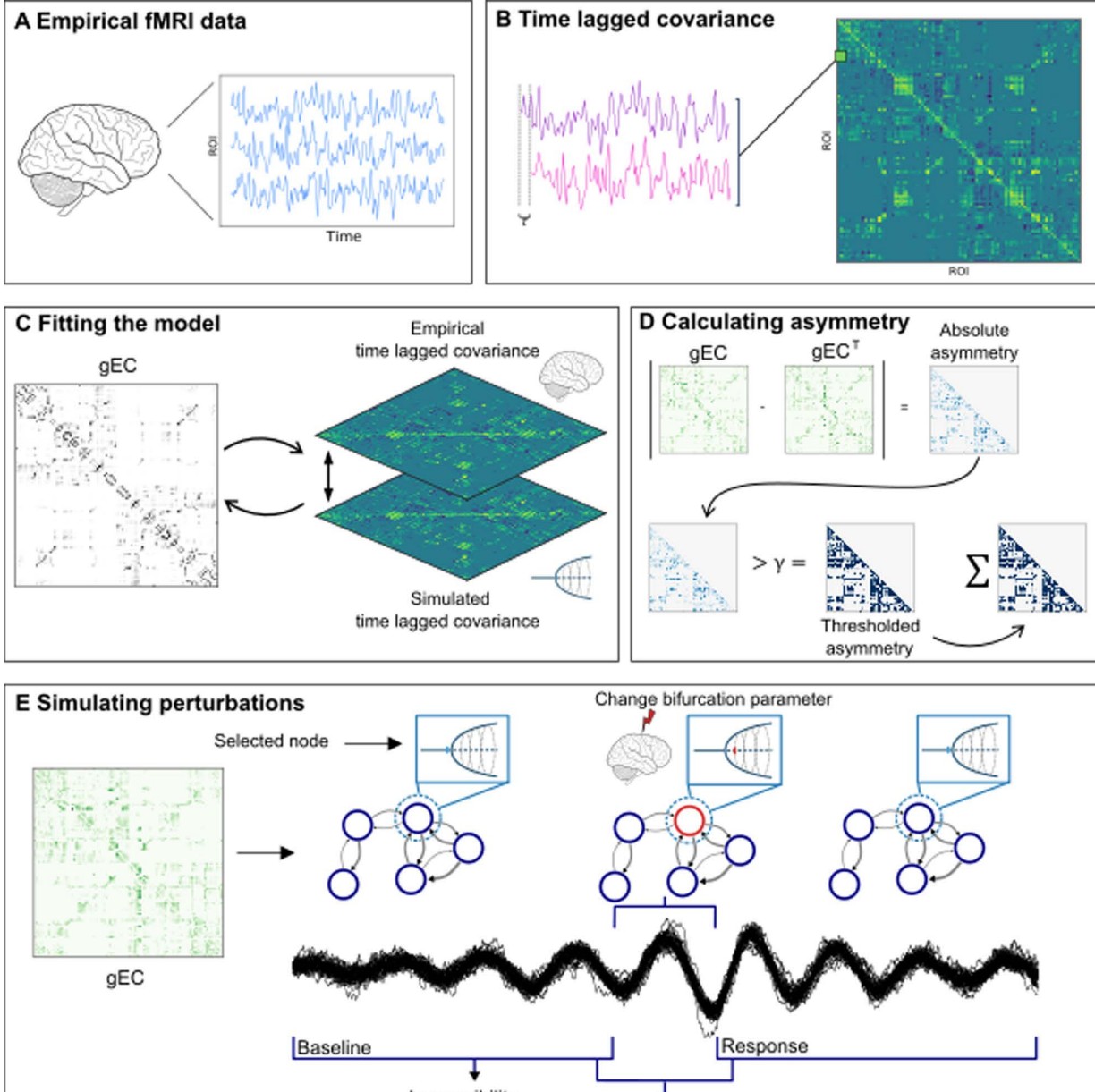

**Fig 1. A schematic overview of the pipeline.** A) fMRI data empirically retrieved from subjects in various states of consciousness, namely control (CNT), Minimally Conscious State (MCS), Unresponsive Wakefulness State (UWS), Wakefulness (W) and deep sleep (N3). B, C) To fit the models, the time lagged covariance is calculated from the empirical data and the simulated data. Through an iterative process in which the generative Effective Connectivity (gEC) is updated, the simulated time lagged covariance will resemble the empirical time lagged covariance as close as possible. This fitting process starts by providing the model with a general Structural Connectivity (see Methods). D) The asymmetry of the gEC is calculated by thresholding the absolute values of the transposed gEC subtracted from its original form, the number of pairs of nodes that exceeded the threshold are summed which gives us the level of asymmetry in the model. E) The gEC is taken as the connectivity matrix in the Hopf model (C in Eqs. 2 and 3), in which each ROI is represented as a Stuart-Landau oscillator in the system. The model is perturbed by changing the value of the bifurcation parameter to a positive value for one of the nodes (a in Eqs. 2 and 3). This perturbation will transcend through the network, and by doing so we get a preperturbational and postperturbational time series, a baseline and a response window respectively, from which we can calculate the State Transition PCI. The irreversibility is calculated from the preperturbational time series, the resting state dynamics of the network. The differences in the gEC for each model will cause the irreversibility and PCI to be different.

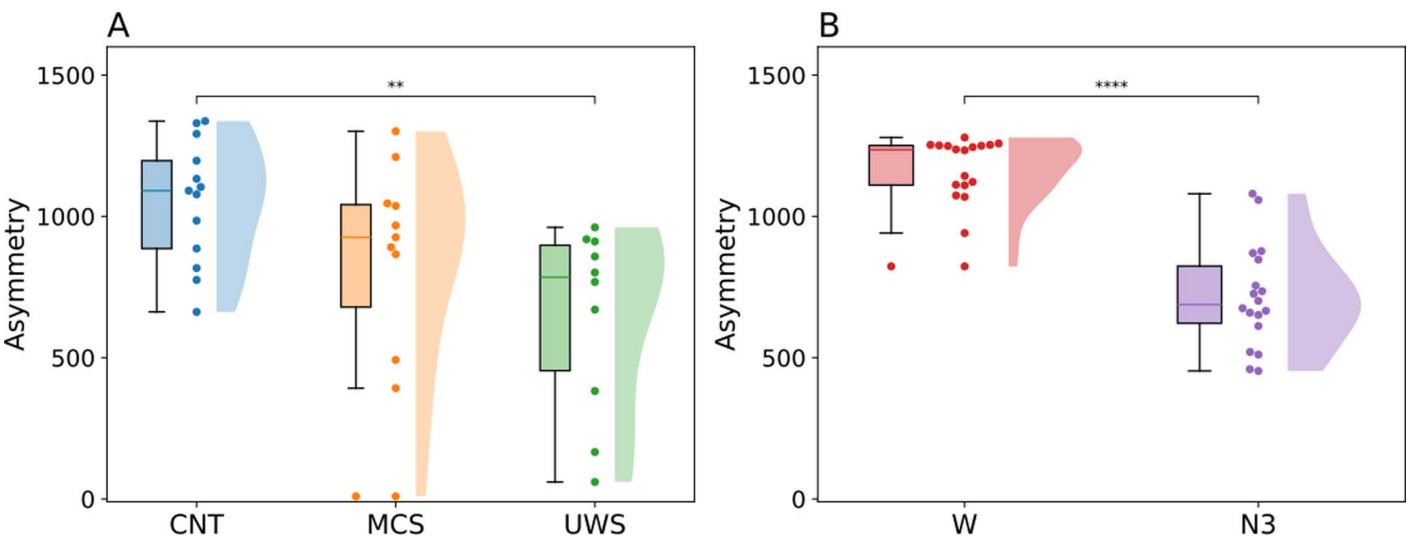

**Fig 2. Comparing the level of asymmetry in the gEC for different states of reduced consciousness.** Asymmetry is calculated for each gEC matrix and thus gives us one value per subject, per state. A) The asymmetry values for the DoC dataset. B) The asymmetry values for the sleep dataset. CNT - control, MCS - Minimally Conscious State, UWS - Unresponsive Wakefulness State, W – wakefulness, N3 – deep sleep state.

subjects (p = 0.0820, Cohen's d = 0.65) no significant differences were found. The distribution of asymmetry values for the MCS does not have a significantly different distribution from the CNT and UWS distributions, but the average does fall between the averages of the CNT and UWS distributions.

For the sleep dataset the difference between the two states W (M = 1161, SD = 125) and N3 (M = 714, SD = 181) was found to be significant (p < 0.001, Cohen's d = 2.65). Taken together, these results suggest that models fitted to subjects residing in normal consciousness tend to have a higher asymmetry in the gEC than those in altered states of consciousness.

### The impact of gEC on the whole-brain nonequilibrium dynamics

As stated before, we used the gEC to inform the connectivity of the model, creating one model per subject ($C_{ij}$ in Eqs. 2 and 3). To look into the level of nonequilibrium of each model, the irreversibility of the preperturbational time series, i.e., the resting state dynamics, was calculated using the INSIDEOUT framework [19]. We ran ten simulations per model for this, yielding 130, 110, 100 and 180 values for subjects in the CNT, MCS, and UWS categories, and the sleep dataset, respectively. As can be seen in Fig 3A all states were significantly different from each other. The difference between CNT (M = 0.0339, SD = 0.0364) and MCS (M = 0.0212, SD = 0.0215) was significant (p = 0.008, Cohen's d = 0.419), as were the differences between MCS and UWS (M = 0.0138, SD = 0.013) distributions (p = 0.002, Cohen's d = 0.412), and between the CNT and UWS distributions (p < 0.001, Cohen's d = 0.704). These results indicate the tendency of irreversibility diminishing when residing in altered states of consciousness, as has been previously shown [20], indicating a state closer to equilibrium. The difference between the irreversibility values of W (M = 0.0607, SD = 0.0588) and N3 (M = 0.0249, SD = 0.0241) were significant as well, where the altered state of consciousness again had a lower level of irreversibility (p < 0.001, Cohen's d = 0.799), which can be observed in Fig 3B.

### Perturbative complexity

To calculate the sfPCI values for the simulations both the preperturbational and the postperturbational time series were used to calculate the sfPCI, based on the State Transition PCI [24]. A simulation was performed for each node for each of

the models, which are depicted individually in the boxplots shown in Fig 4, meaning there are 90 PCI values depicted for each subject. Examples of perturbations for specific nodes can be found in S3 Fig.

Significant differences were found in all three comparisons, after Benjamini-Hochberg correction for multiple comparisons, namely the comparison between the MCS (M = 122, SD = 27) and the UWS (M = 116, SD = 22) group (p < 0.001,

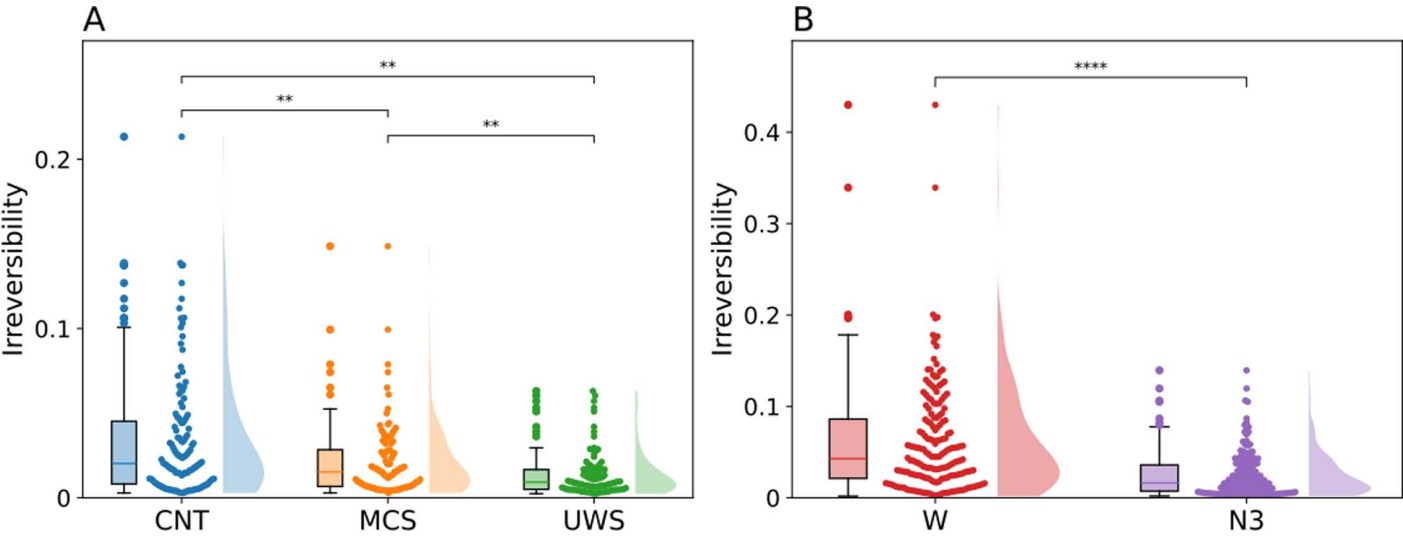

**Fig 3. The underlying gEC generates different whole brain nonequilibrium dynamics.** The INSIDEOUT framework is applied to the simulated time series of each model to calculate the irreversibility. Ten simulations are performed per model. A) The irreversibility values for the DoC dataset. B) The irreversibility values for the sleep dataset. CNT - control, MCS - Minimally Conscious State, UWS - Unresponsive Wakefulness State, W – wakefulness, N3 – deep sleep state.

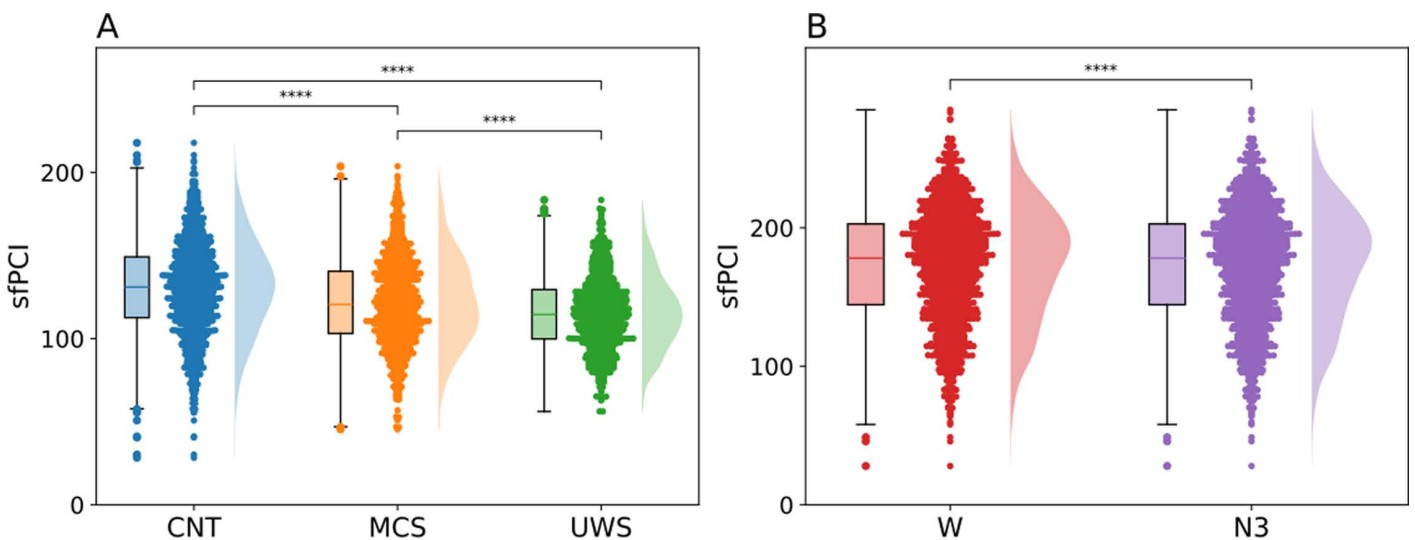

**Fig 4. The underlying gEC generates different PCI values in different levels of consciousness.** The PCI was calculated on simulated perturbations in the model. One node was perturbed per simulation. The figure shows an sfPCI value for each node and each subject. A) The sfPCI values for the DoC dataset. B) The sfPCI values for the sleep dataset. CNT - control, MCS - Minimally Conscious State, UWS - Unresponsive Wakefulness State, W – wakefulness, N3 – deep sleep state.

Cohen's d = 0.250), the comparison between CNT (M = 130, SD = 28) and MCS (p < 0.001, Cohen's d = 0.307), and the comparison between CNT and UWS (p < 0.001, Cohen's d = 0.570). Similar to the irreversibility and asymmetry values the sfPCI has a tendency to be higher in models that were fitted to subjects residing in a normal state of wakeful consciousness. This can be confirmed by looking at Fig 4B where this trend is more pronounced, where the values for the W (M = 173, SD = 41) and N3 (M = 131, SD = 33) states are portrayed (p < 0.001, Cohen's d = 1.129).

**Relation between asymmetry, irreversibility, and sfPCI**

We performed a hundred simulations of perturbations per node per model to calculate the sfPCI. We then averaged the sfPCI values per model, i.e., we have one sfPCI value per model. To calculate the irreversibility, we performed a hundred simulations per model and calculated the average, yielding one irreversibility and one sfPCI value per model. We calculated the Spearman's correlation between the asymmetry and irreversibility, the sfPCI and irreversibility, and the sfPCI and the asymmetry. In Figs 5 and 6 the scatterplots are shown together with their Spearman's correlation values, and the p-value based on a permutation test.

Figs 5 and 6 show that there appears to be a relation between the irreversibility and the asymmetry. This is reflected in the particularly strong correlation values, which were found to be significant in both datasets. For the DoC dataset, this relation is captured by a high correlation (Spearman r = 0.928, p < 0.001), and can similarly be found in the sleep dataset (Spearman r = 0.868, p < 0.001). These correlations across datasets suggest that irreversibility scales with asymmetry, regardless of the type of altered state of consciousness. The relation between sfPCI and irreversibility seems to be a bit more variable, but found to be significant in both datasets after Benjamini-Hochberg correction for multiple comparisons. In the DoC dataset the correlation is weaker (Spearman r = 0.393, p = 0.022), whereas in the sleep dataset the correlation is stronger (Spearman r = 0.686, p < 0.001).

Lastly, sfPCI and asymmetry are found to have a positive significant correlation in both datasets. For the DoC dataset, this correlation is slightly lower (Spearman r = 0.565, p < 0.001) than the strong correlation in the sleep dataset (Spearman

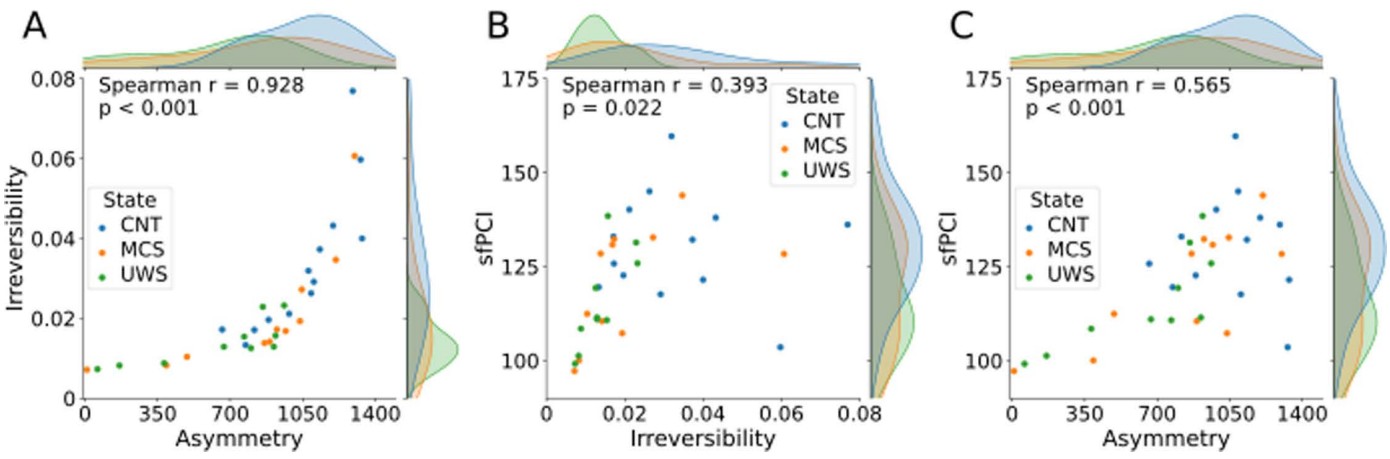

**Fig 5. Relations between the irreversibility, asymmetry, and sfPCI for disorders of consciousness expressed by Spearman's correlation.** Spearman's correlations between the asymmetry, irreversibility, and sfPCI for the simulations from the models fitted to the subjects in the DoC dataset. Hundred simulations were averaged over all ROI's per subject, where the sfPCI values as well as the irreversibility values were averaged per subject. A) The average sfPCI per subject is set out against the corresponding average irreversibility. B) The asymmetry of the gEC is set out against the corresponding average irreversibility. C) The asymmetry of the gEC is set out against the corresponding average sfPCI values. CNT - control, MCS - Minimally Conscious State, UWS - Unresponsive Wakefulness State.

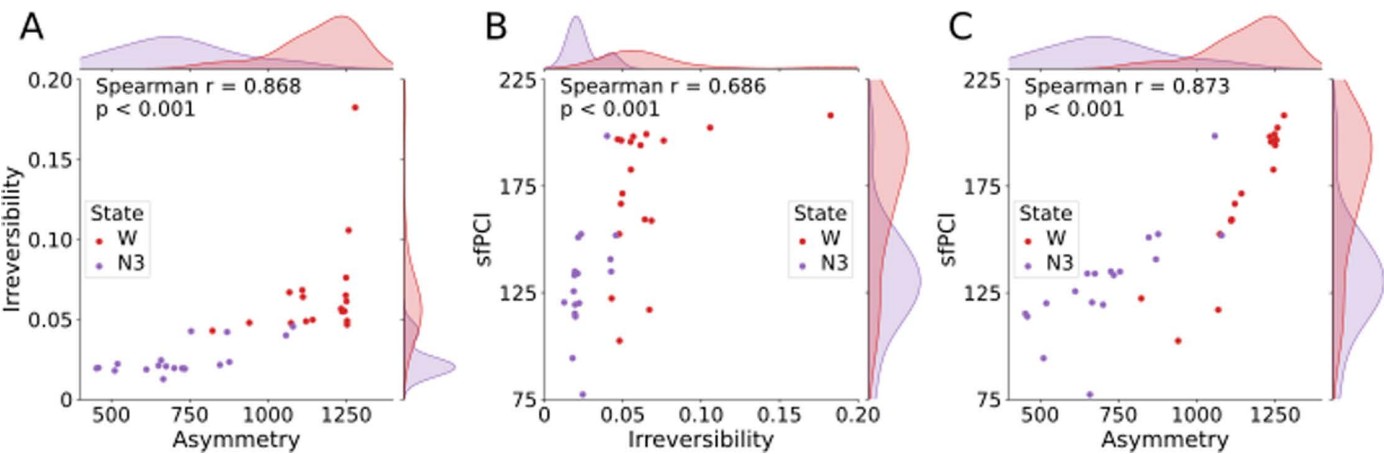

**Fig 6. Relations between the irreversibility, asymmetry and sfPCI for wakefulness and deep sleep expressed by Spearman's correlation.** Spearman's correlations between the asymmetry, irreversibility, and sfPCI, for the simulations from the models fitted to the subjects of the sleep dataset. Hundred simulations were averaged over all ROI's per subject, where the sfPCI values as well as the irreversibility values were averaged per subject. A) The average sfPCI per subject is set out against the corresponding average irreversibility. B) The asymmetry of the gEC is set out against the corresponding average irreversibility. C) The asymmetry of the gEC is set out against the corresponding average sfPCI values. W - wakefulness, N3 - deep sleep state.

r = 0.873, p < 0.001). This indicates that across states of consciousness, higher asymmetry is associated with greater complexity in response to stimuli.

Taken together, our results indicate that higher asymmetry is associated with higher values of sfPCI and irreversibility. This leads us to state that there is a strong and significant correlation between these measures that was not previously reported on in this setup.

Linear mixed effect (LME) models were created and compared to check for group effects based on the states of the subjects for all correlations shown in Figs 5 and 6. The models showed this was not the case, meaning the correlations shown here do not appear to be influenced by the states of the subjects. LME models were created to check for demographic factors as well, including age, etiology of the disordered state, and gender. Since this model includes etiology of the disordered states, it does not include the control subjects. Again, these models showed that the correlations shown here do not appear to be influenced by these demographics.

## Discussion

Our computational modelling approach illuminates how the human brain complexity induced by external stimuli depends on the nonequilibrium dynamics of the brain before being perturbed. Leveraging the thermodynamical description of nonequilibrium in brain dynamics [13,25] and computational whole-brain modelling [26,27], we quantified how the level of nonequilibrium of brain dynamics is related to different states of consciousness and, in turn, how these differences can have an influence on the brain's capability to respond to an external stimulus measured by the sfPCI.

Our goals were twofold. First, we sought to determine whether the level of asymmetry captured in the inferred generative connectivity between brain regions [22] can represent a signature of consciousness, and whether this asymmetry determines the level of nonequilibrium on brain signals. Second, we aimed to determine whether the generative Effective Connectivity (gEC) is also driving the simulated fMRI-based version of the Perturbational Complexity Index of consciousness (sfPCI), establishing a bridge between the nonequilibrium nature of the unperturbed system and the elicited complexity after the perturbation.

Thanks to the recent developments in computational modelling and fMRI data acquisition, we were able to characterise the individual brain dynamics of participants with different altered states of consciousness by fitting their gEC. Importantly, the asymmetry in the gEC is known to break the detailed balance, driving the system out of equilibrium [15,22]. We showed here that higher levels of asymmetry are related to states of normal consciousness. Importantly, we observed that the level of asymmetry decreases both for DoC patients and for participants in deep sleep, indicating that such changes are not specific to a particular altered state of consciousness but are associated with the level of consciousness itself. Previous work has consistently shown the relation between nonequilibrium brain dynamics and altered states of consciousness using empirical data [14,19,20] as well as through whole-brain models [15,21].

It has been widely investigated how the dynamics of the human brain are constrained and supported by the structural connectome [28–31]. In this sense, whole-brain models provided a suitable and successful avenue to investigate how the function of the brain is shaped by its structure [26,29,32,33]. Following the same rationale, we used the generative capabilities of whole-brain models to investigate how the gEC shapes the nonequilibrium of the brain dynamics and how that changes between states of consciousness. Crucially, we found a positive correlation between gEC asymmetry and the level of nonequilibrium in brain dynamics at subject level, suggesting that the broken detailed balance in the generative space determines the level of nonequilibrium at the whole-brain level, as has been demonstrated in other biological systems [34]. Our results also showed that the level of nonequilibrium in brain dynamics obtained with different gEC's significantly decreases as the state of consciousness changes from normal to altered, aligning with empirical results. The results of using the INSIDEOUT framework on the empirical data described in this article can be found in S2 Fig. The tendency for UWS and deep sleep to have lower irreversibility values than CNT and wakefulness can be found, but is more pronounced in our simulated data, most likely due to the data augmentation. Our simulated results also show the distinction between MCS and CNT and between MCS and UWS, not found in the empirical values.

In particular, within the thermodynamics description of brain dynamics, FDT establishes a direct link between the nonequilibrium nature of the system and its response to an external stimulus [15,16,21,35]. This approach provides a necessary theoretical framework for the very influential papers on PCI by Massimini and colleagues. They empirically demonstrated that the complexity after perturbations can be used as biomarkers of consciousness using transcranial magnetic stimulation (TMS) and electroencephalography (EEG) [4–6]. In their work, Massimini and colleagues defined a perturbational complexity index (PCI) that directly measures the amount of information contained in the perturbation-evoked responses by calculating the Lempel-Ziv complexity in space and time of the EEG signals [4]. Following the same principles, they addressed the limitations of the Lempel Ziv PCI and defined the State Transition PCI, using dimensionality reduction and state transition quantification [24]. These PCI measures have been successfully used for separation of brain states in healthy subjects during wakefulness, dreaming, sleep, under different levels of anaesthesia, and in comatose patients [4–6]. Here, we leveraged whole-brain models to compute the State Transition PCI on fMRI-based simulated data by systematically perturbing our brain models *in silico*. While previous works computed the PCI at the group level [36–39], crucially, here we computed an individualized PCI based on the gEC obtained for each participant in each condition. We found that our computational sfPCI confirms the empirical results obtained in previous works [4]. Importantly, as our computational approach allows us to systematically perturb all regions in the brain, we found that the sfPCI decreases as the state of consciousness moves away from the normal wakeful state, independently of the perturbed brain region.

The primary benefit of our proposal compared to earlier empirical methods is that it offers insights into the causal generative mechanisms of complex brain dynamics across various brain states. Crucially, we found that the asymmetry of the gEC underlying the brain dynamics highly correlates with the sfPCI across DoC patients and participants falling asleep, regardless of the perturbed node. This is of particular relevance because, as mentioned in Virmani et al. 2018, an exhaustive manner to perturb the brain of patients is not achievable with the TMS-EEG setup. In this study we show a feasible alternative by using an *in silico* approach [40]. They also state that the PCI is dependent on the current state of the system, in line with our conclusion that the sfPCI is a product of the state of the system, with the notion that the state of

the system is defined by the level of nonequilibrium. In a recent paper of Casarotto et al. 2024 the authors reflect on some shortcomings of the PCI, mentioning that it still needs trained experts to perform this test, and not all patients are eligible for TMS procedures, limiting its usefulness [41]. They also note several strict criteria to which this TMS-EEG needs to be upheld. Following the line of thought that the PCI is determined by the level of nonequilibrium *in silico*, this could be an indication that the same holds for the empirical PCI and that the TMS-EEG setup could be redundant. This was previously concluded in the work describing the ECI through their ECI[awa] (ECI in awareness component) based on empirical resting state data [11], yielding similar predictive accuracies as the PCI itself. However, in this work we have emphasised the level of nonequilibrium being the necessary information in resting-state data.

The link between the empirically performed perturbations and PCI values and their simulated equivalents is beyond the scope of this work and should be investigated in future research. Previous studies using empirical data have consistently shown that altered states of consciousness, lower arousal and/or lower awareness, are associated with lower PCI values and reduced irreversibility levels [8,20,24]. This leads us to the hypothesis that there would be a strong correlation between the empirical PCI and the metrics presented here. Importantly, future work should focus on investigating this relationship based on empirical measures of non-equilibrium dynamics and FDT approaches [21,42] with the empirical PCI.

Several theories of consciousness have been created over the years, amongst which some well-known theories exist such as the Global Neuronal Workspace Theory (GNWT) [43], the Integrated Information Theory (IIT) [44], and the Temporo-spatial Theory of Consciousness (TTC) [45–47]. Since the PCI stems from IIT, our results fall in line with their ideas around compressibility of neural activity, sharing similar results with their empirical setup where normal consciousness results in less compressible activity. The TTC not being bound by integration, but relying on temporospatial dynamics as the basis of consciousness, draws an interesting line to our results presented here. They discuss the unperturbed activity as the context in which a perturbation and integration can take place [48]. Here the gEC creates the basis and the level of nonequilibrium can be seen as the context. The irreversibility can be quantified, creating a context for the sfPCI.

The work presented here should be considered within certain limitations. Firstly, all models were fitted from the same structural connectivity matrix, gathered from healthy subjects. Considering the DoC patients can have significant brain damage, this might influence the way these subjects are represented in our models. Secondly, the AAL parcellation has 90 regions, which is rather low considering that many parcellations used in computational neuroscience nowadays can often have 200 up to >1000 regions. Perturbational research might be more refined when using these parcellations, but computationally speaking (and coming from an environmentally sustainable point point of view), these works are much heavier. Thirdly, the number of subjects in the datasets, in particular the DoC dataset, is rather low. This restricts us from dividing the MCS patient group into MCS- and MCS+ and gives us a less powerful conclusion altogether. Lastly, there is no direct link made between the empirical PCI and the sfPCI presented in this work. The translation made between our computational setup and a biological setting are therefore assumptions, based within a larger story of connected empirical and computational works. The current research is restricted in several ways that could be interesting to follow up on in future research. A larger dataset could help in investigating how other factors come into play, such as the time since injury, and a distinction between MCS+ and MCS-. A follow-up study of that could be to see how our models predict the prognosis of these patients. Furthermore, a connection between the computational and empirical PCI's would be an important step in giving our claims more foundation.

Taken together, these findings indicate that the broken detailed balance of the brain dynamics, reflected in the asymmetry of the underlying gEC, can be used as a model-based biomarker of consciousness. In turn, this asymmetry lies at the root of the difference in nonequilibrium and sfPCI, establishing a direct link between the nonequilibrium of the unperturbed system and its responsiveness to external stimuli *in silico*. Additionally, it has potential to aid in reducing the need for costly experiments and enhance statistical reliability.

## Materials and methods

### Ethics statement

This research was approved by the local ethics committee Comité de Protection des Personnes Ile de France 1 (Paris, France) under the code 'Recherche en soins courants' (NEURODOC protocol, no. 2013-A01385-40). The patients' relatives gave their formal written informed consent for their familiar to participate, and all investigations were performed according to the Declaration of Helsinki and the French regulations.

### Disorders of consciousness data

We used fMRI data from 21 patients with DoC's. 10 of those patients were diagnosed with UWS (3 female, mean age 38.02±std of 17.02) and the other 11 patients were diagnosed with MCS (4 female, mean age 40.85±std of 15.04). No distinction was made between MCS+ and MCS-. fMRI data was collected from an additional 13 healthy control subjects (7 female, mean age 42.54±std of 13.64). This dataset comes from a larger dataset previously described in Escrichs et al. 2022 [49]. Trained clinicians conducted the clinical assessment and CRS-R scoring to determine the patients' level of consciousness. Patients were diagnosed with MCS if they showed some behaviour that could be indicative of awareness, such as visual pursuit, orientation to pain, or reproducible command following. Patients were diagnosed with UWS if they showed signs of arousal (through opening their eyes) without any signs of awareness (never showing non-reflex voluntary movements).

The fMRI data for this dataset were acquired with a 3T General Electric Signa System. T2*-weighted whole-brain resting state images were captured with a gradient-echo EPI sequence using axial orientation (200 volumes, 48 slices, slice thickness: 3 mm, TR/TE: 2400 ms/30 ms, voxel size: 3.4375 × 3.4375 × 3.4375 mm, flip angle: 90°, FOV: 220 mm$^2$). An anatomical volume was obtained using a T1-weighted MPRAGE sequence in the same acquisition session (154 slices, slice thickness: 1.2 mm, TR/TE: 7.112 ms/3.084 ms, voxel size: 1 × 1 × 1 mm, flip angle: 15°).

Pre-processing of the resting state data was performed using FSL (http://fsl.fmrib.ox.ac.uk/fsl) as described previously [49]. Resting state fMRI was computed using MELODIC (multivariate exploratory linear optimized decomposition into independent components) [50]. Steps included discarding the first five volumes, motion correction using MCFLIRT [51], brain extraction tool (BET) [52], spatial smoothing with 5 mm FWHM Gaussian kernel, rigid-body registration, high pass filter cutoff at 100.0 s, and single-session independent component analysis (ICA) with automatic dimensionality estimation. Lesion-driven artefacts (for patients) and noise components were regressed out independently for each subject using FIX (FMRIB's ICA-based X-noiseifier) [53]. The time series were extracted in the AAL parcellation [23].

### Sleep and wakefulness data

Written informed consent was obtained, and the study was approved by the ethics committee of the Faculty of Medicine at the Goethe University of Frankfurt, Germany. The data used in this work comes from a larger database. The participants included in this work were those that reached all four polysomnographic sleep stages.

We used fMRI data from 18 subjects residing in both wakefulness (W) and deep sleep (N3), previously described in [15,54]. The data was acquired using a 3T system (Siemens Trio, Erlangen, Germany) with settings: 1505 volumes of T2*-weighted echo planar images with a repetition time (TR) of 2.08 seconds, and an echo time of 30 ms; matrix 64 x 64, voxel size 3 x 3 x 2 mm$^3$, distance factor 50%, field of view (FOV) 192 mm2.

The EPI data were realigned, normalised to MNI space, and spatially smoothed using a Gaussian kernel of 8 mm$^3$ FWHM in SPM8 (http://www.fil.ion.ucl.ac.uk/spm/). Spatial downsampling was then performed to a 4 x 4 x 4 mm$^3$ resolution. From the simultaneously recorded ECG and respiration, cardiac- and respiratory-induced noise components were estimated using the RETROICOR method [55], which were regressed out of the signals together with motion parameters. The data were temporally band-pass filtered in the range 0.008-0.08 Hz using a sixth-order Butterworth filter. The time series were extracted in the AAL parcellation [23].

Simultaneously to acquiring the fMRI data, polysomnography (PSG) was performed by recording EEG, EMG, ECG, EOG, pulse oximetry, and respiration. EEG was recorded using a cap (modified BrainCapMR, Easycap, Herrsching, Germany) with 30 channels, where the FCz electrode was used as reference. The sampling rate of the EEG was 5 kHz, a low-pass filter was applied at 250 Hz and a high-pass filter at 0.016 Hz. MRI and pulse artefact correction were applied based on the average artefact subtraction method [27] in Vision Analyzer2 (Brain Products, Germany), followed by objective (CBC parameters, Vision Analyzer) ICA-based rejection of residual artefact-laden components after subtracting the average artefact, resulting in EEG with a sampling rate of 250 Hz. EMG was collected with chin and tibial derivations, while the ECG and EOG were recorded bipolarly at a sampling rate of 5 kHz with a low-pass filter at 1 kHz. Pulse oximetry was collected using the Trio scanner, at a sampling rate of 50 Hz, and respiration was recorded with MR-compatible devices (BrainAmp MR+, BrainAmp ExG; Brain Products, Gilching, Germany). Participants were instructed to lie still in the scanner with their eyes closed and relax. Sleep classification was performed by a sleep expert based on the EEG recordings in accordance with the American Academy of Sleep Medicine criteria (2007) [56].

## Structural connectivity

We used the SC consisting of 90 ROI's as presented in a previous study, for a more detailed description please refer to [57], described in the section 'DWI data collection and processing'. This dataset, and the SC gathered from it, are separate from the previously described datasets on DoC and sleep. Briefly, a SC matrix was obtained for each subject (n = 16) by using tractography algorithms to Diffusion Tensor Imaging, following a previously described methodology [58]. Participants were healthy and right-handed, and recruited online at Aarhus University in Denmark. Subjects that had a current diagnosis or a history of psychiatric or neurological disorders were not included. The connection between regions $C_{ij}$ was calculated as the proportion of sampled fibres in all voxels in region i that reach any voxel in region j. DTI does not portray directionality of these fibres, therefore the average was taken between $C_{ij}$ and $C_{ji}$ as the undirected connection between regions i and j. The average was taken over the 16 subjects to portray a SC matrix representing a healthy connectivity. In this work, we use this SC as a starting point from where the fitting procedure can begin. In other words, this SC is used as the base from which the fitting procedure starts. It is also used as a mask during fitting, determining where connections can exist in the model, and where not, regardless of the weight of those connections.

## Statistical testing and multiple comparison correction

To test whether two distributions are significantly distinct from one another we used a permutation based ranksum t-test with 10,000 permutations. To correct for multiple comparisons, we used a Benjamini-Hochberg procedure, where the significance threshold ($\alpha = 0.05$) is adjusted for each p-value based on their rank in order of magnitude. The threshold is then determined by dividing the p-value by the number of tests performed and multiplying by its rank. For example, if there are three tests performed these significance thresholds are based on $\alpha = 0.05/3 = 0.0167$, multiplied by their rank. For rank 1, rank 2, and rank 3 these would be $\alpha = 0.0167$, $\alpha = 0.033$, and $\alpha = 0.05$, respectively.

To test whether two metrics are correlated we used Spearman's correlation to check for monotonic correlations and a non-parametric permutation-based test with 10,000 permutations. Again, significance thresholds for the p-values were corrected through a Benjamini-Hochberg procedure.

Linear mixed effect (LME) models were created to check whether these correlations were not influenced by the states of the subjects. LME models were created with and without the states as random effects, and were checked to be significantly different or not from one another through a likelihood ratio test (LRT). For the LME models to check the influence of demographic factors, the ages of the subjects were binned to create a discretized variable for the random effect, while the other factors were already discrete. This model included as random effects the age, etiology of the disordered state, gender, and the diagnosis of the patients. It did not include the control subjects, since the etiology of the disordered state was included as one of the random effects.

## Effect size

To calculate and interpret the effect size of the difference between two distributions we used Cohen's d effect size [59]. To describe effect size we use the descriptors small, medium, and large, described by Sullivan and Fein [59] by using cutoff values of 0.2, 0.5, and 0.8, respectively. We adjusted the effect sizes where needed when dealing with smaller sample sizes (N < 50) to correct for possible instability in small sample sizes.

## Whole-brain modelling

The simulations are done by using the Hopf model, a model of coupled Stuart Landau oscillators. These oscillators are described by the normal form of a supercritical Hopf bifurcation that represent the different ROI's. The ROIs, i.e., the nodes in the model, are connected through a Structural Connectivity (SC) derived from empirical data. The bifurcation parameter of the Hopf bifurcation is set to a value close to 0 so that the oscillator portrays the oscillatory and noisy behaviour that is characteristic for the brain [26]. In Eq. 1 the dynamics of one oscillator are depicted in its uncoupled form, where $z_j$ represents the dynamics as a complex value (with $z_j = x_j + y_j$). In this equation $\omega$ represents the intrinsic frequency of the node, calculated from the fMRI data as the averaged peak frequency and bandpass filtered between 0.04 and 0.07 Hz, and $a$ the bifurcation parameter which was set at -0.02:

$$\frac{dz_j}{dt} = (a + i\omega)z_j - z_j \left|z_j\right|^2 \tag{1}$$

As previously mentioned, the oscillators are connected through an infrastructure represented by an SC, depicted as $C_{ij}$ in Eqs. 2 and 3:

$$\frac{dx_j}{dt} = \left(a - x_j^2 - y_j^2\right)x_j - \omega_j y_j + G\Sigma_i C_{ij}(x_i - x_j) + \beta\eta_j(t) \tag{2}$$

$$\frac{dy_j}{dt} = \left(a - x_j^2 - y_j^2\right)y_j + \omega_j x + G\Sigma_i C_{ij}(y_i - y_j) + \beta\eta_j(t) \tag{3}$$

Similar as per Eq. 1 $a$ represents the bifurcation parameter and $\omega$ represents the intrinsic frequency of the node, and $\eta$ noise. The global coupling factor is represented by G and in this work it is kept at a value of 1 for all models. For a more detailed description of the Hopf model, please refer to Deco et al. 2017 [26].

## Generative effective connectivity

Each subject's data was used to fit a generative Effective Connectivity (gEC) to create personalised models by using the linear approximation of the Hopf model [60], depicted in Fig 1C. All data were band-pass filtered between 0.04-0.07 Hz before fitting. A model is fitted to the empirical time-lagged covariances, depicted in Eq. 6, by iteratively adjusting the values of the connections in $C_{ij}$ which are used in Eqs. 2 and 3 to simulate time series. The fitting procedure starts by providing the model with the weights of the SC (obtained through DTI as mentioned above). The model is run and the simulated time-lagged covariances are calculated and compared to their empirical equivalent values. The weights of the model are then adjusted so that the simulated and empirical time-lagged covariances are as close as possible until the error between the two has stagnated (see S1 Fig). The personalisation of the models was done by first creating grouped models. The time-lagged covariance matrix of each subject was calculated and averaged per group, yielding five grouped models: three grouped models for the DoC dataset (CNT, MCS, UWS) and two grouped models for the sleep dataset (W, N3). From there on each personalised model was created by taking the gEC of their respective grouped model as an initial step for the fitting process. Since the empirical time-lagged covariances come from the

individual data per state, this results in personalised weights for each of the models, representing each subject in each state. An SC mask is used, meaning that all gECs in the end have connections in the same constellation but not with the same weights.

## Asymmetry

The asymmetry of a model was calculated by looking at the number of pairs that were asymmetrically connected in the gEC. An intuitive overview of this metric is shown in Fig 1D. The transposed gEC was subtracted from its original form, yielding a symmetric matrix. The absolute values of this symmetric matrix were binarized by using a threshold, 0 for values below the threshold and 1 for values above the threshold, as per Eq. 4. The asymmetry value depicts the number of pairs of nodes, and thus the sum of the lower triangle was taken as the number of pairs that were connected sufficiently asymmetric to survive the thresholding. Mathematically this can be seen as summing matrix A yielded in Eq. 4 and divided by two as per Eq. 5.

$$A = \begin{cases} 1 \ for \ |a_{ij} - a_{ji}| \geq \gamma \\ 0 \ for \ |a_{ij} - a_{ji}| < \gamma \end{cases}$$ (4)

$$N = \frac{\Sigma A}{2}$$ (5)

The threshold was generalized for all models and set so that all asymmetry values were at least non-zero. In other words, all gECs should have at least one pair of nodes that was connected asymmetrically enough to surpass the threshold, which was found to be at a value of 0.12.

## Insideout

To compute the irreversibility of the resting state dynamics of the models we used the INSIDEOUT framework [19]. This framework looks at the temporal asymmetry of time series by comparing the time shifted covariance matrices of the forward and the reversed time series, as shown in Eq. 6:

$$c_{forward}(\Delta t) = < x(t), y(t + \Delta t) >$$ (6)

By looking at the distance of these covariance matrices, a value is calculated to portray how high the level of irreversibility is in the system, this depicts the arrow of time of the system and represents the level of non-equilibrium. For a multidimensional system this equation looks like:

$$FS_{forward,ij} = 1/2 log \left( 1 - < x_i(t), x_j(t + \Delta t) >^2 \right)$$ (7)

$$FS_{reversed,ij} = 1/2 log \left( 1 - < x_i^{(r)}(t), x_j^{(r)}(t + \Delta t) >^2 \right)$$ (8)

Where the irreversibility is calculated as the mean of the absolute squares of the elements of the difference between $FS_{forward}$ and $FS_{reversed}$:

$$I(T) = \left| \left| FS_{forward}(T) - FS_{reversed}(T) \right| \right|_2$$ (9)

For a more detailed description of this framework, please refer to Deco et al. 2022 [19].

## PCI simulated

In this work we made use of a simulated version of the PCI in which we relied on fMRI data, called the simulated fMRI-based PCI (sfPCI). To introduce perturbations in the model we made use of the Hopf model's bifurcation characteristics. By changing the value of the bifurcation parameter to a positive value a node is pushed into an oscillatory state. By doing so, the oscillatory activity of this node will affect the other nodes based on their connectedness as dictated by the gEC. The preperturbational time series is used as a baseline and the postperturbational time series is used as a response window as defined by the methodology of the State Transition PCI. A principal component analysis is done on the response window and by using a singular value decomposition the baseline is expressed in these principal components as well. The principal components that make up 99% of the response are the ones that are used in the analysis of the number of state transitions. The PCI is the summed difference in number of state transitions between the baseline and the response window expressed in principal components. For a more detailed description of the State Transition PCI please refer to Comolatti et al. 2019 [24].

## Supporting information

**S1 Fig. An example curve is shown of the fitting procedure for the gEC.** In an iterative process the time-lagged covariances of the simulated time series are compared to those of the empirically retrieved time series. In the case of a positive error, i.e., the empirical value is higher than the simulated value, the weight of the gEC of that connection is increased to create a stronger connection and consequently a higher value in the time-lagged covariance. For a negative error the process is the same but reversed, the weight of the connection is decreased. The error shown in this graph is the error over all time-lagged covariances in the matrix of this model.
(TIFF)

**S2 Fig. The INSIDEOUT framework is applied to the empirically retrieved time series of each subject in to calculate the irreversibility, resulting in one value per subject per state.** A) The irreversibility values for the DoC dataset. B) The irreversibility values for the sleep dataset. CNT - control, MCS - Minimally Conscious State, UWS - Unresponsive Wakefulness State, W – wakefulness, N3 – deep sleep state.
(TIFF)

**S3 Fig. The sfPCI was calculated on simulated perturbations.** A perturbation of each node was simulated 100 times, i.e., each boxplot contains a number of datapoints equal to a 100 times the number of subjects in that category. In this figure, examples are depicted per node, showing that this trend where PCI is higher in CNT and W than in MCS, UWS and N3, is present regardless of the node perturbed. Less nodes are portrayed for the DoC dataset for visualization purposes. A) The sfPCI values for the DoC dataset. B) The sfPCI values for the sleep dataset. CNT - control, MCS - Minimally Conscious State, UWS - Unresponsive Wakefulness State, W – wakefulness, N3 – deep sleep state.
(TIFF)

## Author contributions

**Conceptualization:** Wiep Stikvoort, Gustavo Deco, Yonatan Sanz Perl.

**Data curation:** Iván Mindlin.

**Formal analysis:** Wiep Stikvoort, Yonatan Sanz Perl.

**Funding acquisition:** Gustavo Deco.

**Investigation:** Wiep Stikvoort.

**Methodology:** Wiep Stikvoort, Gustavo Deco, Yonatan Sanz Perl.

**Resources:** Iván Mindlin, Anira Escrichs, Jacobo D. Sitt.

**Software:** Wiep Stikvoort, Eider Pérez-Ordoyo, Gustavo Deco, Yonatan Sanz Perl.

**Supervision:** Gustavo Deco, Yonatan Sanz Perl.

**Visualization:** Wiep Stikvoort.

**Writing – original draft:** Wiep Stikvoort, Yonatan Sanz Perl.

**Writing – review & editing:** Wiep Stikvoort, Eider Pérez-Ordoyo, Iván Mindlin, Anira Escrichs, Jacobo D. Sitt, Morten L. Kringelbach, Gustavo Deco, Yonatan Sanz Perl.

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
