## [Decision Letter · Decision Letter 0]

PCOMPBIOL-D-24-02115

Nonequilibrium brain dynamics elicited as the origin of perturbative complexity

PLOS Computational Biology

Dear Dr. Stikvoort,

Thank you for submitting your manuscript to PLOS Computational Biology. After careful consideration, we feel that it has merit but does not fully meet PLOS Computational Biology's publication criteria as it currently stands. Therefore, we invite you to submit a revised version of the manuscript that addresses the points raised during the review process.

Particularly important points raised include increasing the clarity of the analyses and motivation for each of the steps, toning down the language of the results to reflect the findings, and discussing the limitations of the current work.

Please submit your revised manuscript within 60 days Mar 16 2025 11:59PM. If you will need more time than this to complete your revisions, please reply to this message or contact the journal office at ploscompbiol@plos.org. Please include the following items when submitting your revised manuscript:

We look forward to receiving your revised manuscript.

Kind regards,

Amy Kuceyeski

Academic Editor

PLOS Computational Biology

Daniele Marinazzo

Section Editor

PLOS Computational Biology

**Journal Requirements:**

At this stage, the following Authors/Authors require contributions: Wiep Stikvoort, Eider Pérez-Ordoyo, Iván Mindlin, Anira Escrichs, Jacobo D. Sitt, Morten L. Kringelbach, Gustavo Deco, and Yonatan Sanz Perl. Please ensure that the full contributions of each author are acknowledged in the "Add/Edit/Remove Authors" section of our submission form.

Potential Copyright Issues:

i) Figures 1A, 1C, and 1E. Please confirm whether you drew the images / clip-art within the figure panels by hand. If you did not draw the images, please provide (a) a link to the source of the images or icons and their license / terms of use; or (b) written permission from the copyright holder to publish the images or icons under our CC BY 4.0 license. Alternatively, you may replace the images with open source alternatives. See these open source resources you may use to replace images / clip-art:

5) In the online submission form, you indicated that "The disorder of consciousness datasets contain information from a clinical population and are not publicly available due to constraints imposed by the approved ethics protocol. Data can be shared upon request to the authors." All PLOS journals now require all data underlying the findings described in their manuscript to be freely available to other researchers, either

1. In a public repository

2. Within the manuscript itself

3. Uploaded as supplementary information.

For studies involving third-party data, we encourage authors to share any data specific to their analyses that they can legally distribute. PLOS recognizes, however, that authors may be using third-party data they do not have the rights to share. When third-party data cannot be publicly shared, authors must provide all information necessary for interested researchers to apply to gain access to the data. For more information, see:

https://journals.plos.org/ploscompbiol/s/data-availability#loc-acceptable-data-access-restrictions

4) All necessary contact information others would need to apply to gain access to the data.

1) State what role the funders took in the study. If the funders had no role in your study, please state: "The funders had no role in study design, data collection and analysis, decision to publish, or preparation of the manuscript.".

**Reviewers' comments:**

Reviewer's Responses to Questions

Reviewer #1: This paper is one by of the leading groups in the field. They converge resting state fMRI in reduced consciousness (sleep, UWS) and converge them with their computational model to investigate the underlying thermodynamics. They observe that reduced states of consciousness show lower asymmetry in their effective connectivity, lower time irreversibility, and lower complexity. This is a well done paper which therefore only raises a few questions.

1. There seems to be a considerable discrepancy between authors summary and abstract with the PCI only being mentioned in the former but not the latter, that leaves me confused….

2. The introduction then starts right away with the PCI…as I Understand this is about PCI in the modelling rather than with TMS…

3. They may want to better explain their notion of asymmetry in the introduction..not clear to me..and also the link from PCI to thermodynamics..

4. I am not sure: when converging the data with the model – do they take a structural connectivity matrix (DT) from the healthy usbjects, from a healthy data base or from the patients themselves? Even in the method this is not fully clear ot me, sorry

5. This should be clarified in the result part, I have not yet look at the method part..the fitting procedure of model and data is illustrated in figure 1 but not fully to me from that figure and the result text…

6. I am confused about the concept of generative effective connectivity…what is the difference to the standard concept of effective connectivity? Just that is now done in the model? Or does it show a substantial difference to the effective C in the data?

7. The data in figure 3 do not look fully convincing to me.,.lots of outliers and large overlaps between HC and patients…….is that based on and the result of the personalizing the model of the individual gEC data?

8. They may want to show some empirical data of the group comparison…this will tell us how much the model is really better than the empirical data as the authros seem to claim

9. They need to mention and cite some other literature on consciousness like the Temporo-spatial approach by the group around Northoff which also focuses on brain dynamics in disorders of consciousness although in slightly distinct ways….cite their recent papers and book…

10. And discuss some of the theories of Consciousness like iIT, GNWT and TTC….given that they apply one key measure of the IIT, the pCI…/.

Reviewer #2: PCOMPBIOL-D-24-02115: Nonequilibrium brain dynamics elicited as the origin of perturbative complexity

Comments to the authors

In this study, the authors apparently reanalyzed previously acquired data to created personalized whole-brain models of effective connectivity fitted to rsfMRI data of control subjects, and subjects in deep sleep and others with MCS or UWS. They then determined the out-of-equilibrium nature of brain dynamics by evaluating the level of asymmetry and time irreversibility in each model and compared it with a simulated complexity as elicited by a perturbation (mimicking TMS in empirical data). They found that the level of asymmetry, irreversibility, and complexity at a resting state predicts the complexity of a response to perturbation.

This is an interesting study. The paper is well written, clear, convincing, and straightforward. However, I would need a little bit more precision to perfectly understand what was actually done, and to correct a few weaknesses that appear in the paper. My concerns are detailed hereafter.

- The authors refer to previous studies having described the PCI as indicative of the complexity of brain dynamics. Another index has been described, the ECI, which can be calculated on EEG data without the need for TMS. Two types of ECI exist, one evaluating the level of arousal (ECIaro), and the other the level of information segregation and integration (ECIawa). I would be interested in knowing how the authors relate their findings to this ECI. It might be interesting to discuss this.

- In the dataset obtained from patients with disorders of consciousness, it is not clear to me whether the distinction between MCS+ and MCS- patients was done.

- The authors should be cautious regarding the terminology they use. At several places, they mention ‘reduced consciousness’ or ‘higher levels of consciousness’. To me, this terminology is too vague, because it is very difficult to quantify the degree of reduction or enhancement. I think the terms ‘altered consciousness’, ‘absence of consciousness’ or ‘normal consciousness’ would be much more appropriate.

- The authors do not discuss the limitations of their study. One limitation is that they used data acquired from other studies and reanalyzed them. This should be made clear, and associated limitations (sample size was not dedicated to the present study, there might have been a risk of type II statistical error, particularly regarding the results reported in Figure 2A). There are probably other limitations related to their analysis pipeline, which should be addressed (see also the other concerns of this reviewer).

- The brain was parcellated into 90 ROIs. This is a quite low number (some studies report parcellation into more the 1000 ROIs). Please explain this choice.

- The authors mention having used the Benjamini-Hochberg correction for multiple comparisons. This correction is usually used to avoid identifying too many false positives (which is the case with the highly conservative Bonferroni correction). Please be more precise regarding the alpha threshold that was chosen for the BH correction, and the rationale behind this choice.

- In the paragraph entitled ‘gEC asymmetry characterizes different states of reduced consciousness’, the authors first report results obtained in DoC, then in sleep, then again in DoC. Please reorder the way results are presented.

- Please describe more clearly how subjects were chosen in the respective datasets (all subjects? Matched controls?).

- Please briefly describe how the EEG data (I presume acquired during fMRI acquisition in the sleep dataset) were cleaned before being assessed by expert for sleep stage classification.

- I personally do not understand the role of DTI in the analysis that was performed for this study. Please explain.

- Regarding the Cohen’s d analysis, I would like to know what the authors considered a small, medium, or large effect size. This would allow having a better idea of the risk of type II statistical error.

Reviewer #3: I thank the editor for the opportunity to review this manuscript regarding the nature of how brain dynamics is related to perturbative complexity in various states of consciousness. The manuscript, combining empirical and model-based approaches, shows that level of non-equilibrium dynamics at rest can predict how the modelled brain would react to a perturbation. This is an interesting idea with potential clinical applications. However, the use of the stPCI in fMRI data to disentangle states of consciousness has not been validated, which is important given that the temporal and spatial resolution of EEG (to which the stPCI is usually applied) is very different from fMRI. Furthermore, the manuscript poses bold statements regarding the causal relation between nonequilibrium brain dynamics and perturbative complexity, while merely correlations are used to support this claim and potentially many unknown or untested lurking variables could mediate the reported correlation.

Before giving some more concrete suggestions, I would like to this this opportunity to share some more general observations about how this work is presented. First, I’d like to ask the authors to revisit the statement that that clinicians must rely on behaviour rather than quantifiable metrics. Clinical assessments such as the CRS-R are designed to be quantifiable, rather than the clinical consensus diagnosis which is indeed associated with high levels of misdiagnosis (i.e., about 40%). If the authors refer to patients possessing covert consciousness instead, then it is true that neuroimaging assessments are useful (and recommended in the EU and USA) to complement the diagnosis. However, typically these auxiliary assessments are being validated on the behavioural assessment of patients – hence they are important and still the gold standard. I would therefore like to ask the authors for a clarification on this point, or to take caution when suggesting that neuroimaging-based measures would be “better” to objectify levels of consciousness than clinical assessment.

Related to this, I believe that the implications of the current work are more modest than portrayed in the current version of the manuscript. Statements like “Through our models we demonstrated that the asymmetry in the generative underlying connections is causing the emergent nonequilibrium dynamics and, at the same time, the differences in the complexity elicited by perturbations measured with PCI.”, “Following the line of thought that the PCI is determined by the level of nonequilibrium, indicates that the TMS-EEG setup could be redundant.” And “Additionally, it could reduce the need for costly experiments, enhance statistical reliability, and minimize potential ethical issues.” are in my opinion problematic, as (1) the use of the stPCI has not been validated with real data. Though this limitation is mentioned in lines 314-320, its consequences are not reflected in the interpretation of the results nor discussion. (2) the causality of these relationships has not been proven and can thus still only be assumed. (3) It is questionable which ethical issues would be resolved by utilizing this model as there is a large overlap between the different groups and more generally, I wonder how ethical it is to replace an in-vivo experiment with a (compared to the actual brain) simplified model of it. Though I do see merit in these types of computational approaches, I believe it is important to also be mindful about their limitations.

More specifically, yet related to the points raised above, I have the following questions:

-The original/ empirical PCI is well-known for its diagnostic precision. In the current work it seems that there is a large overlap between the diagnostic groups and states of consciousness. The reader might wonder if this is true, or if the overlap seems so large because the stimulation of all regions is presented in one figure. Do different figures show different PCI’s? Based on the paragraph starting on line 30, it seems that the PCI is invariant to the perturbed region, but I fail to find the data supporting that statement.

-Related to this, it might be better to not use “PCI” when describing the results, but rather something along the lines of “model-based fMRI PCI”

-Do the authors have an explanation for why irreversibility and the PCI seems to be higher in the awake HC group than in the DoC HC group?

-I understand that no real PCI data of these patients is available to cross-check the fMRI model-based PCI. However, I believe that patients visiting this clinical center receive several other assessments (e.g., FDG-PET and EEG) that are used to refine the clinical diagnosis. If that is true, where there any covert conscious patients in this cohort, and how did the model’s prediction behave in these patients?

-Regarding the relations between the irreversibility, asymmetry, and PCI, I am also interested to know if there would be a group effect. The authors have tested this using an LME with the group as random effect. I wonder why they did also including other clinical/demographic variables as time since injury, age etc? I understand that the sample size is modest, yet some preliminary investigation in that direction could be worthwhile. Also, the description of the LME approach should probably be moved to the methods section.

-I do not understand which 16 datasets were used to create the average SC. The HC used in the HC data consists of 13, the HC from the sleep data consist of 18, then there are the patients, but it is unclear which data is used (and why only a selection is used?) to create the SC.

-Related to the previous point, if the SC is based in the HC alone, how can the authors infer that it is representative for DoC patients as well? One could for example imagine that DoC patients show relatively more local connectivity than HC, but this would then not (or less so) be reflected in the CS, and consequently not be considered in the analysis. This could perhaps lead to an underestimation of the different metrics (and an overestimation of group differences).

-I recommend the authors to report the group mean and SD for each of the metrics presented in the results section.

-Terms like irreversibility could be introduced in more detail so that the paper becomes accessible to a broader readership.

**Have the authors made all data and (if applicable) computational code underlying the findings in their manuscript fully available?**

Reviewer #1: Yes

Reviewer #2: **No: ** DoC dataset is not publicly available. Otherwise, the authors all data and code readily available.

Reviewer #3: **No: ** A part of the dataset is made available, however, the patient data is not publicly available. The code is made available.

PLOS authors have the option to publish the peer review history of their article (what does this mean? ). If published, this will include your full peer review and any attached files.

**Do you want your identity to be public for this peer review?** For information about this choice, including consent withdrawal, please see our Privacy Policy .

Reviewer #1: No

Reviewer #2: **Yes: ** Vincent Bonhomme

Reviewer #3: No

**Figure resubmission:**
---

## [Decision Letter · Decision Letter 1]

Dear Ms Stikvoort,

We are pleased to inform you that your manuscript 'Nonequilibrium brain dynamics elicited as the origin of perturbative complexity' has been provisionally accepted for publication in PLOS Computational Biology.

Best regards,

Amy Kuceyeski

Academic Editor

PLOS Computational Biology

Daniele Marinazzo

Section Editor

PLOS Computational Biology

Reviewer's Responses to Questions

**Comments to the Authors:**

Reviewer #1: all quetsions well addressed

Reviewer #2: The authors have adequately and very nicely addressed my concerns. Thank you for giving me the opportunity to review this work of excellent quality.

Reviewer #3: Thank you for addressing my concerns, I have no further feedback.

**Have the authors made all data and (if applicable) computational code underlying the findings in their manuscript fully available?**

Reviewer #1: Yes

Reviewer #2: Yes

Reviewer #3: **No: ** see previous review

PLOS authors have the option to publish the peer review history of their article (what does this mean? ). If published, this will include your full peer review and any attached files.

**Do you want your identity to be public for this peer review?** For information about this choice, including consent withdrawal, please see our Privacy Policy .

Reviewer #1: No

Reviewer #2: **Yes: ** Vincent Bonhomme

Reviewer #3: No

---

## [Editor Report · Acceptance letter]

PCOMPBIOL-D-24-02115R1

Nonequilibrium brain dynamics elicited as the origin of perturbative complexity

Dear Dr Stikvoort,

I am pleased to inform you that your manuscript has been formally accepted for publication in PLOS Computational Biology. Your manuscript is now with our production department and you will be notified of the publication date in due course.

With kind regards,

Zsofia Freund
